# Inferring Latent Velocities from Weather Radar Data using Gaussian Processes

**Rico Angell**
University of Massachusetts Amherst
rangell@cs.umass.edu

**Daniel Sheldon**
University of Massachusetts Amherst
sheldon@cs.umass.edu

## Abstract

Archived data from the US network of weather radars hold detailed information about bird migration over the last 25 years, including very high-resolution partial measurements of velocity. Historically, most of this spatial resolution is discarded and velocities are summarized at a very small number of locations due to modeling and algorithmic limitations. This paper presents a Gaussian process (GP) model to reconstruct high-resolution full velocity fields across the entire US. The GP faithfully models all aspects of the problem in a single joint framework, including spatially random velocities, partial velocity measurements, station-specific geometries, measurement noise, and an ambiguity known as aliasing. We develop fast inference algorithms based on the FFT; to do so, we employ a creative use of Laplace's method to sidestep the fact that the kernel of the joint process is non-stationary.

## 1 Introduction

Archived data from the US network of weather radars hold valuable information about atmospheric phenomona across the US for over 25 years [1]. Although these radars were designed to monitor weather, they also detect flying animals such as birds, bats, and insects [2]. The information contained in the archive is critical to understanding phenomena ranging from extreme weather to bird migration [3–5].

This paper is concerned with using radar to measure velocity, with the primary goal of gathering detailed information about bird migration. Radar is the most comprehensive source of information about this difficult-to-study phenomenon [5–8], but, historically, most information has gone largely unused due to the sheer size of the data and the difficulty of interpreting it automatically. Recently, analytical advances including machine learning [9, 10] are enabling scientists to begin to conduct larger scale studies [5, 7, 11]. Radar measurements of bird migration density, direction, and speed are important for understanding the biology of bird migration and to guide conservation [11–15]. Machine learning methods to automate the detailed interpretation of radar data will allow scientists to answer questions at the scale of the entire continent and over more than two decades.

Doppler radars measure the rate at which objects approach or depart the radar, which gives partial information about their velocity. By making certain smoothness assumptions, it is possible to reconstruct full velocity vectors [9, 16]. However, current methods are limited by rigid smoothness assumptions and summarize all velocity information down to 143 points across the US (the locations of the radar stations) even though the original data has on the order of half a billion measurements for one nationwide snapshot.

The goal of this paper is to develop a comprehensive, principled, probabilistic model, together with fast algorithms, to reconstruct spatially detailed velocity fields across the US. There are three critical challenges. First, radars only measure *radial velocity*, the component of velocity in the direction of

the radar beam, so the full velocity is underdetermined. Second, the measured radial velocity may be *aliased*, which means it is only known up to an additive constant. Third, measurements are tied to station-specific geometry, so it is not clear how to combine data from many stations, for example to fill in gaps in coverage between stations (e.g., see Figure 1(d)). Prior research has primarily addressed these challenges separately, and has been unable to combine information from many radars to reconstruct detailed velocity fields.

Our first contribution is a joint Gaussian process (GP) to simultaneously model the radial velocity measurements from all radar stations. While it is natural to model the velocity field itself as a GP, it is not obvious how to model the collection of all station-specific measurements as a GP. We start by positing a GP on latent velocity vectors, and then derive a GP on the measurements such that the station-specific geometry is encoded in the kernel function.

Our second contribution is a suite of fast algorithms for inference in this GP, which allows it to scale to very large data sets. We leverage fast FFT-based algorithms for GP kernel operations for points on a regular grid [17–19]. However, these require a stationary kernel, which due to the station-specific geometry, ours is not. We show how to achieve the same speed benefits by using Laplace's method (for *exact* inference) so that fast kernel operations can be performed in the space of latent velocities, where the kernel is stationary. Finally, we show how to model aliasing directly within the GP framework by employing a *wrapped normal likelihood* [9, 20]; this fits seamlessly into our fast approach using Laplace's method.

The result is a first-of-its-kind probabilistic model that jointly models all aspects of the data generation and measurement process; it accepts as input the raw radial velocity measurements, and outputs smooth reconstructed velocity fields.

## 2 Background and Problem Definition

**Radar Basics.** The US network of weather radars, known as "NEXRAD" radars, consists of 143 radars in the continental US. Each conducts a *volume scan* or *scan* every 6 to 10 minutes, during which is rotates its antenna 360 degrees around a fixed vertical axis (one "sweep") at increasing elevation angles. The result of one scan is a set of raster data products in three-dimensional polar coordinates corresponding to this scanning strategy. One measurement corresponds to a particular antenna position (azimuth and elevation angle) and range; the corresponding volume of atmosphere at this position in the polar grid is called a *sample volume*.

NEXRAD radars collect up to six different data products. For our purposes the most important are *reflectivity* and *radial velocity*. *Reflectivity* measures the density of objects, specifically, the total cross-sectional area of objects in a sample volume that reflect radio waves back to the radar. *Radial velocity* is the rate at which objects in a sample volume approach or depart the radar, which is measured by analyzing the frequency shift of reflected radio waves (the "Doppler effect"). Radial velocity is illustrated in Figure 1(a). For any given sample volume, radial velocity gives only *partial* velocity information: the projection of the actual velocity onto a unit vector in the direction of the radar beam. However, if the actual velocity field is smooth, we can often make good inferences about the full velocity. Figure 1(b) shows example radial velocity information measured from the KBGM radar in Binghamton, NY on the night of September 11, 2010, during which there was heavy bird migration. Objects approaching the radar have negative radial velocities (green), and objects departing the radar have positive radial velocities (red). We can infer from the overall pattern that objects (in this case, migrating birds) are moving relatively uniformly from northeast to southwest.

**Velocity Model.** To make inferences of the type in Figure 1(b) we need to simultaneously reason about spatial properties of the velocity field and the measurement geometry. To set up this type of analysis, for the $i$th sample volume within the domain of one radar station, let $\mathbf{a}_i$ be the unit vector in the direction from the radar station to the sample volume. This is given by $\mathbf{a}_i = (\cos \phi_i \cos \rho_i, \sin \phi_i \cos \rho_i, \sin \rho_i)$ where $\phi_i$ and $\rho_i$ are the azimuth and elevation angles, respectively. Let $\mathbf{z}_i = (u_i, v_i, w_i)$ be the actual, unobserved, velocity vector. Then the *radial* velocity is $\mathbf{a}_i^T \mathbf{z}_i$, and the *measured* radial velocity is $y_i = \mathbf{a}_i^T \mathbf{z}_i + \epsilon_i$. Here, $\epsilon_i \sim \mathcal{N}(0, \sigma^2)$ is zero-mean Gaussian noise that plays the dual role of modeling measurement error and deviations from whatever prior model is chosen for the set of all $\mathbf{z}_i$. For example, in the *uniform velocity model* [16], velocities are assumed to be constant-valued within fixed height bins above ground level within the domain of

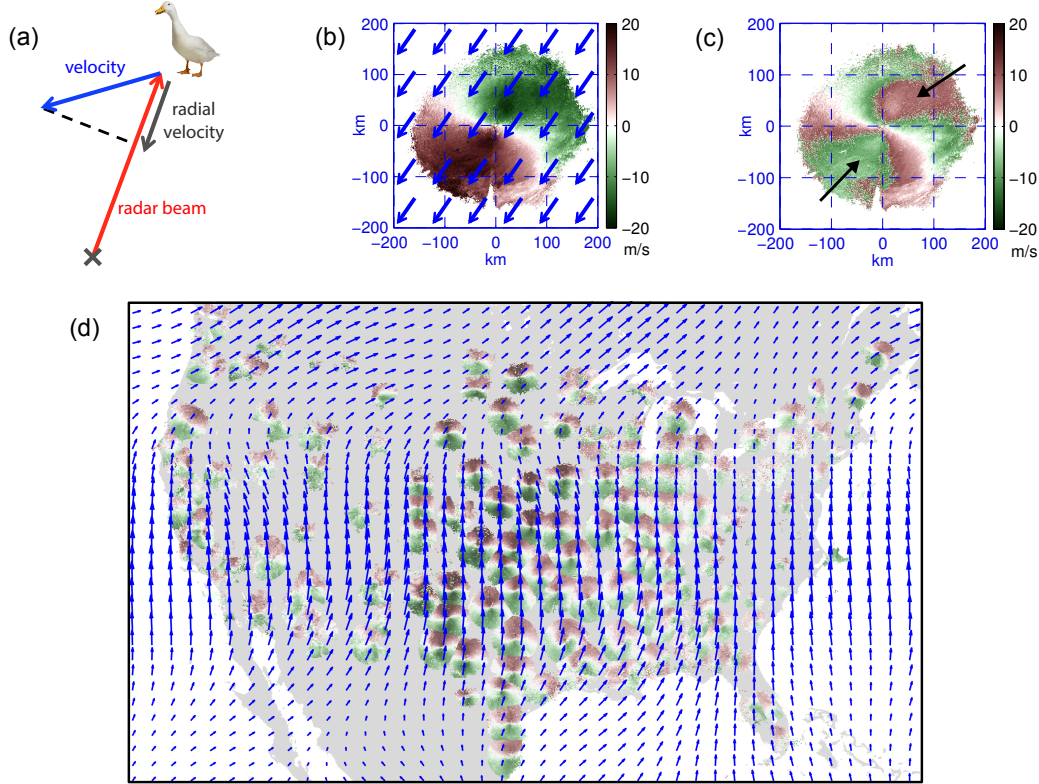

Figure 1: Illustration of key concepts: (a) schematic of radial velocity measurement, (b) radial velocity in the vicinity of Binghmaton, NY radar station during bird migration event on Sep 11, 2010, (c) aliased radial velocity, (d) a nationwide mosiac of raw radial velocity data is not easily interpretable, but we can extract a velocity field from this inforation (arrows). See text for explanation.

one radar station, which is a very rigid uniformity assumption. Reported values for the noise standard deviation are $\sigma \in [2, 6]\,\mathrm{ms}^{-1}$ for birds, and $\sigma < 2\,\mathrm{ms}^{-1}$ for precipitation [7].

**Aliasing.** Aliasing complicates the interpretation of radial velocity data. Due to the sampling frequency of the radars, radial velocities can only be resolved up to the *Nyquist* velocity $V_{\max}$, which depends on the operating mode of the radar. If the magnitude of the true radial velocity $r_i = \mathbf{a}_i^T \mathbf{z}_i$ exceeds $V_{\max}$, then the measurement will be *aliased*. The aliasing operation is mathematically equivalent to the modulus operation: for any real number $r$, define the *aliased measurement* of $r$ to be $\bar{r} := r \mod 2V_{\max}$, with the convention that $\bar{r}$ lies in the interval $[-V_{\max}, V_{\max}]$ instead of $[0, 2V_{\max}]$. The values $\bar{r} + 2kV_{\max}, k \in \mathbb{N}$ will all result in the same aliased measurement, and are called *aliases*. Effectively, this means that radial velocities will "wrap around" at $\pm V_{\max}$. For example, Figure 1(c) shows the same data as Figure 1(b), but before aliasing errors have been corrected. In this example $V_{\max} = 11\mathrm{ms}^{-1}$. We see that that fastest approaching birds in the northeast quadrant appear to be departing (red), instead of approaching (dark green).

**Multiple Radar Stations.** The interpretation of radial velocity is station-specific. Figure 1(d) shows a nationwide mosaic of radial velocity from individual stations, overlaid by a velocity field. The mosaic is very difficult to interpret, due to abrupt changes at the boundaries between station coverage areas. Thus, although we are very accustomed to seeing nationwide composites of radar reflectivity, radial velocity data is not presented or analyzed in this way. This is the main problem we seek to remedy in this work, by reconstructing velocity fields of the type overlaid on Figure 1(d).

**Related Work.** The *uniform velocity model* [16], described above, makes a strong spatial unformity assumption to reconstruct velocities at different heights in the immediate vicinity of one radar station. Variants of this method are known as *velocity volume profiling* (VVP) or *velocity-azimuthal display* (VAD). The uniformity assumptions prevent these algorithms from reconstructing spatially varying velocity fields or combining information from multiple radars. *Multi-Doppler* methods combine

measurements from two or more radars to reconstruct full velocity vectors at points within the overlap of their domains [16, 21, 22]. No spatial smoothness assumptions are made. Full velocity fields can be reconstructed, but only within the overlap of radar domains. *Dealiasing* is the process of correcting aliasing errors to guess the true radial velocity, usually by making smoothness assumptions or using some external information [23]. Almost all previous work treats the different analytical challenges (reconstruction from spatial cues, multiple stations, dealiasing) separately; a few methods combine dealiasing with VVP or multi-Doppler methods [9, 24, 25]. Our method extends *all* of these methods into a single, elegant, joint probabilistic model.

# 3  Modeling Latent Velocities

In this section, we present our joint probabilistic model for radial velocity measurements and latent velocities. We begin by considering the problem in the absence of aliasing, and come back to it in Section 4.

**Likelihood in the absence of aliasing.**  Let $\mathcal{O}_i$ be the set of stations that measure radial velocities at location $\mathbf{x}_i$. The likelihood of a single radial velocity measurement $y_{ij}$, in the absence of aliasing, given the latent velocity $\mathbf{z}_i$ and the radial axis $\mathbf{a}_{ij}$, is Gaussian around the perfect radial velocity measurement of the ground-truth latent velocity

$$p(y_{ij}|\mathbf{z}_i;\mathbf{x}_i) = \mathcal{N}(y_{ij};\mathbf{a}_{ij}^T\mathbf{z}_i,\sigma^2). \tag{1}$$

The observed radial velocity measurements are conditionally independent given the latent velocities, so the joint likelihood factorizes completely

$$p(\mathbf{y}|\mathbf{z};\mathbf{x}) = \prod_i \prod_{j\in\mathcal{O}_i} p(y_{ij}|\mathbf{z}_i;\mathbf{x}_i) = \prod_i \prod_{j\in\mathcal{O}_i} \mathcal{N}(y_{ij};\mathbf{a}_{ij}^T\mathbf{z}_i,\sigma^2). \tag{2}$$

**GP prior.**  We model the latent velocity field as a vector-valued GP. The GP prior has a zero-valued mean function and a modified squared exponential kernel. Since the GP is vector-valued, the output of the kernel function is a $3 \times 3$ matrix of the following form.

$$\kappa_\theta(\mathbf{x}_i,\mathbf{x}_j) = \mathrm{diag}\left(\exp\left(\frac{-d_\alpha(\mathbf{x}_i,\mathbf{x}_j)}{2\beta_u}\right), \exp\left(\frac{-d_\alpha(\mathbf{x}_i,\mathbf{x}_j)}{2\beta_v}\right), \exp\left(\frac{-d_\alpha(\mathbf{x}_i,\mathbf{x}_j)}{2\beta_w}\right)\right) \tag{3}$$

$$d_\alpha(\mathbf{x}_i,\mathbf{x}_j) = \alpha_1(\mathbf{x}_{i,1}-\mathbf{x}_{j,1})^2 + \alpha_2(\mathbf{x}_{i,2}-\mathbf{x}_{j,2})^2 + \alpha_3(\mathbf{x}_{i,3}-\mathbf{x}_{j,3})^2 \tag{4}$$

The hyperparameters $\theta = [\alpha,\beta]$ are the length scales which control the uniformity of the latent velocity field.

**Covariance between measurements.**  Our approach to inferring the latent velocities relies on the ability to jointly model the radial velocity measurements with the latent velocities. In order to accomplish this, we need to have a covariance function relating radial velocity measurements. Intuitively this seems problematic, since the radial velocity measurements not only depend on the location of the measurement, but also the location of the station making the measurement. As it turns out, applying definitions and the process by which radial velocity measurements are made gives the following elegant covariance function.

$$\mathrm{Cov}(y_{ij},y_{i'j'}) = \mathbb{E}[y_{ij}y_{i'j'}] = \mathbf{a}_{ij}^T\mathbb{E}[\mathbf{z}_i\mathbf{z}_{i'}^T]\mathbf{a}_{i'j'} = \mathbf{a}_{ij}^T\kappa_\theta(\mathbf{x}_i,\mathbf{x}_{i'})\mathbf{a}_{i'j'}$$

Observe that this covariance function is not stationary, since it relies on the locations of the stations from which the measurements were made.

**Joint modeling measurements and latent velocities.**  The joint probability distribution between the radial velocity measurements and the latent velocities is

$$p(\mathbf{y},\mathbf{z};\mathbf{x}) = p(\mathbf{y}|\mathbf{z};\mathbf{x})p(\mathbf{z};\mathbf{x}). \tag{5}$$

Since both the likelihood and prior are Gaussian, the joint is also Gaussian. All we need to do to fully specify the joint distribution is to solve for the first two moments of the joint. The joint mean is clearly zero. Let $\mathbf{q}^T = [\mathbf{z}^T\ \mathbf{y}^T]$, let $A = \mathrm{diag}\left(\{\mathbf{a}_{ij}^T\,|\,\forall i,j\in\mathcal{O}_i\}\right) \in \mathbb{R}^{3n\times n}$ be the matrix defined

---

**Algorithm 1** Efficient Inference using Laplace's Method

---

1: **procedure** INFERLATENTVELOCITIES
2:     Initialize $\nu^{(0)}$ randomly                                        $\triangleright \nu^{(0)} = K^{-1}\mathbf{z}^{(0)}$
3:     Initialize $\Delta\nu = \infty$
4:     **while** $|\Delta\nu| > \tau$ **do**                                   $\triangleright \tau$ is some user-defined threshold
5:         Compute $b = W\mathbf{z}_k + \nabla l(\mathbf{z}_k)$
6:         Compute $\gamma = (W^{-1} + K)^{-1}Kb$ using the conjugate gradient method
7:         Let $\Delta\nu = b - \gamma - \nu^{(k)}$
8:         Set $\nu^{(k+1)} = \nu^{(k)} + \eta\Delta\nu$                       $\triangleright$ Use Brent's method to do a line search for $\eta$
9:     **return** $\mathbf{z}^* = K\nu^*$

---

so that $\mathbf{y} \sim \mathcal{N}(A\mathbf{z}, \sigma^2 I)$, and let $K$ be the prior covariance matrix. The covariance of the joint is as follows

$$\mathbb{E}[\mathbf{q}\mathbf{q}^T] = \left[\begin{bmatrix} \mathbf{z} \\ \mathbf{y} \end{bmatrix} \begin{bmatrix} \mathbf{z}^T & \mathbf{y}^T \end{bmatrix}\right] = \left[\begin{array}{c|c} K & KA^T \\ \hline AK^T & AKA^T + \sigma^2 I \end{array}\right] \tag{6}$$

Hence, the joint distribution is

$$p(\mathbf{y}, \mathbf{z}; \mathbf{x}) = \mathcal{N}\left(\begin{bmatrix} \mathbf{z} \\ \mathbf{y} \end{bmatrix}; \mathbf{0}, \left[\begin{array}{c|c} K & KA^T \\ \hline AK^T & AKA^T + \sigma^2 I \end{array}\right]\right). \tag{7}$$

**Naive Exact Inference.**   Given this joint distribution, we can perform exact inference via Gaussian conditioning. The posterior mean is

$$\mathbb{E}[\mathbf{z}|\mathbf{y}; \mathbf{x}] = KA^T(AKA^T + \sigma^2 I)^{-1}\mathbf{y}. \tag{8}$$

We can also predict directly at locations $\tilde{\mathbf{z}}$ other than those where measurements were made using the cross-covariance matrix $\tilde{K}$ between the locations where measurements were made and prediction locations:

$$\mathbb{E}[\tilde{\mathbf{z}}|\mathbf{y}; \mathbf{x}] = \tilde{K}A^T(AKA^T + \sigma^2 I)^{-1}\mathbf{y}. \tag{9}$$

This method of inference is not scalable since it has cubic time complexity and quadratic space complexity in the number of measurements.

# 4   Efficient Inference

In this section, we discuss how we can perform efficient exact inference despite the lack of a stationary kernel.

## 4.1   Laplace's Method for *Exact* Inference

In order to make inference tractable, we would like to use fast FFT-based methods such as SKI and KISS-GP [18], but unfortunately these methods require the kernel to be stationary. To overcome having a non-stationary kernel, we apply Laplace's method [26]. This is conventionally for *approximate* inference when the likelihood is not Gaussian, but we use it to be able to utilize fast kernel operations for the latent GP, which is stationary, and the method will still be exact. Laplace's method replaces one-shot matrix inversion based inference with an iterative algorithm where the most complicated operation is kernel-vector multiplication. If we pick locations to observe radial velocity measurements on a grid $\Omega$, we can perform the matrix-vector multiplication $K\mathbf{s}$, for an arbitrary vector $\mathbf{s}$, in $O(n \log n)$ time, where $n = |\Omega|$.

The exact inference procedure we employ is presented in Algorithm 1. Laplace's method iteratively optimizes $\log p(\mathbf{z}|\mathbf{y}; \mathbf{x})$ by optimizing the second-order Taylor expansion around the current iterate of $\mathbf{z}$ via an auxiliary variable $\nu = K^{-1}\mathbf{z}$. Let $l(\mathbf{z}) = \log p(\mathbf{z}|\mathbf{y}; \mathbf{x})$ be the log likelihood function, $\nabla l(\mathbf{z})$ be the gradient of the log likelihood, and $W = -\nabla^2 l(\mathbf{z})$ be the negative Hessian. The most challenging operation to make efficient is Line 6 of Algorithm 1. We use the conjugate gradient method to iteratively compute $\gamma$. The upshot is that we only need to be able to efficiently compute

$W^{-1}$, multiply $W^{-1}$ times arbitrary vectors, and multiply $K$ times arbitrary vectors. $W$ is block diagonal with $3 \times 3$ blocks, which makes for linear time matrix-vector multiplication and inversion. The only other bottleneck for both speed and storage is the kernel matrix.

## 4.2 Using Grid Structure for Fast Matrix-Vector Multiplication

In this section, we detail how we can perform efficient kernel-vector multiplication by exploiting the special structure of the kernel matrix following techniques presented by Wilson [27]. To accomplish this we need to choose the measurements to use as observations from an evenly spaced grid. In most cases, we will not have measurements for all grid points, so we use pseudo-observations to enable the use of grid-based methods.

### 4.2.1 Missing Observations

Given $\Omega$ to be the set of grid locations where we would like to have radial velocity measurements, let $\hat{\Omega}$ and $\tilde{\Omega}$ be the locations where we have and do not have radial velocity measurements, respectively. For all grid locations $\mathbf{x}_i \in \tilde{\Omega}$, we sample a pseudo radial velocity measurement $y_i \sim \mathcal{N}(0, \epsilon^{-1})$, for some small $\epsilon$. This implies the following joint log likelihood:

$$l(\mathbf{z}) = \sum_i \left( \mathbb{1}[\mathbf{x}_i \in \tilde{\Omega}] \log \mathcal{N}(y_i; 0, \epsilon^{-1}) + \mathbb{1}[\mathbf{x}_i \in \hat{\Omega}] \left( \sum_{j \in \mathcal{O}_i} \log \mathcal{N}(y_{ij}; \mathbf{a}_{ij}^T \mathbf{z}_i, \sigma^2) \right) \right). \quad (10)$$

### 4.2.2 Kronecker-Toeplitz Structure

The latent GP can be decomposed into three independent GP's – namely, over the $u$, $v$, and $w$ components of the latent velocities, respectively. Let $K_u$, $K_v$, and $K_w$ be kernel matrices for each of these GP's, respectively, and all have shape $n \times n$. When performing the multiplication $K\mathbf{s}$, we decompose $\mathbf{s}$ into it's $u$, $v$, and $w$ component sub-vectors denoted $\mathbf{s}_u$, $\mathbf{s}_v$, and $\mathbf{s}_w$, respectively. Then, we perform each of the multiplications $K_u\mathbf{s}_u$, $K_v\mathbf{s}_v$, and $K_w\mathbf{s}_w$, and recombine the results to get $K\mathbf{s}$. All of these three multiplications are similar since $K_u$, $K_v$, and $K_w$ all have the same structure.

We use $K_u$ as an example and follow the method proposed by Wilson [27]. $K_v$ and $K_w$ follow the same form. $K_u$ decomposes into the Kronecker product $K_{u,1} \otimes K_{u,2} \otimes K_{u,3}$, where $K_{u,1}$, $K_{u,2}$, and $K_{u,3}$ are all Toeplitz, since $K_u$ is stationary. $K_{u,1}$ has shape $n_1 \times n_1$, $K_{u,2}$ has shape $n_2 \times n_2$, and $K_{u,3}$ has shape $n_3 \times n_3$ where $n_1$, $n_2$, and $n_3$ are the dimensions of the grid, respectively. Hence, $n = n_1 n_2 n_3$. Let $\mathbf{S}_u$ be the $n_1 \times n_2 \times n_3$ tensor formed by reshaping $\mathbf{s}_u$ to match the grid dimensions. Then

$$K_u \mathbf{s}_u = \left( \bigotimes_{i=1}^3 K_{u,i} \right) \mathbf{s}_u = \text{vec}\left( \mathbf{S}_u \times_1 K_{u,1} \times_2 K_{u,2} \times_3 K_{u,3} \right).$$

Here, the operation $\mathbf{T} \times_i M_i$ denotes the $i$-mode product of the tensor $\mathbf{T} \in \mathbb{R}^{n_1 \times n_2 \times n_3}$ and matrix $M_i \in \mathbb{R}^{n_i \times n_i}$. The result is another tensor $\mathbf{T}'$ with the same dimensions. It is computed by first reshaping $\mathbf{T}$ into a matrix $\mathbf{T}_{(i)}$ of size $n_i \times \prod_{j \neq i} n_j$, then computing the matrix product $M_i \mathbf{T}_{(i)}$, and finally reshaping the result back into an $n_1 \times n_2 \times n_3$ tensor — see [28] for details. In our case, since each matrix multiplication is between a Toeplitz matrix $K_{u,i}$ and a matrix $\mathbf{T}_{(i)}$ with $n$ entries, it can be done in $O(n \log n)$ time using the FFT [29]. Therefore, the overall running time is also $O(n \log n)$.

## 4.3 Handling Aliased Data

In this section, we extend our model to handle aliased radial velocity measurements. Recall that aliasing means that radial velocities are only known up to an additive multiple of twice the Nyquist velocity $V_{\max}$, which varies by operating mode of the radar. Conditions favorable for bird migration often correspond to low values of $V_{\max}$ and exacerbate aliasing problems.

To accommodate aliasing, we change the likelihood to model the aliasing process using a *wrapped normal likelihood* [20]:

$$p(y_{ij}|\mathbf{z}_i; \mathbf{x}_i) = \mathcal{N}_w(y_{ij}|\mathbf{a}_{ij}^T \mathbf{z}_i, \sigma^2) = \sum_{k=-\infty}^{\infty} \mathcal{N}(y_{ij} + 2kV_{\max,j}; \mathbf{a}_{ij}^T \mathbf{z}_i, \sigma^2) \quad (11)$$

This is simply the marginal density of all aliases of $y_{ij}$. The infinite sum cannot be computed analytically, so we approximate it with a finite number of aliases, $\ell$, which is known to perform well [9, 30, 31].

$$p(y_{ij}|\mathbf{z}_i;\mathbf{x}_i) \approx \mathcal{N}_w^\ell(y_{ij}|\mathbf{a}_{ij}^T\mathbf{z}_i, \sigma^2) = \sum_{k=-\ell}^{\ell} \mathcal{N}(\overline{y_{ij} - \mathbf{a}_{ij}^T\mathbf{z}_i} + 2kV_{\max,j}; 0, \sigma^2) \qquad (12)$$

Recall that $\bar{r}$ aliases $r$ to the interval $[-V_{\max}, V_{\max}]$, so the sum on the right-hand side is over the $2\ell + 1$ aliases of $y_{ij}$ that are closest to the predicted value $\mathbf{a}_{ij}^T\mathbf{z}_i$. Since our efficient inference method only relies on the likelihood only through its gradient and Hessian, we can simply plug these new functions into the algorithm presented in Algorithm 1. Observe that this likelihood is no longer Gaussian, and thus we are no longer performing exact inference using Laplace's method.

## 5  Experiments

In this section, we present the results from experiments to evaluate the effectiveness of the method we presented in the previous section. The first two experiments analyze data scans from 13 radar stations from the northeast US on the night of September 11, 2010. In all experiments, hyperparameters are fixed at values chosen through preliminary experiments to match the expected smoothness of the data, so that the RMSE between inferred radial velocities and raw measurements match values from velocity models used in prior research [7, 9].

**Comparison of inference methods.**  First, we compare our fast inference method against the naive inference method. In our experiments we first resample data from all radar stations onto a fixed resolution grid. Each grid point has zero or more observations from different radar stations. The naive method operates only on the actual observations $m$, and its running time is $O(m^3)$. Our grid-based method operates on all $n$ grid points, and its per-iteration running time is $O(n \log n)$. To tractably perform naive inference we must subsample the $m$ observations even further. We consider a range of different sizes both for the base grid and the subsampled data set for the naive method.

Figure 2 shows the time vs. error for six different methods. The data set consists of radar scans from 13 radar stations from the northeast US on the night of September 11, 2010, and, for this test, is preprocessed to eliminate aliasing errors [9]. Error is measured by first inferring the full velocity vector for each observation and then projecting it using the station-specific geometry to compute the RMSE between the predicted and observed *radial* velocities. To fairly compare RMSE values across the six methods, the naive method must predict values for all observations, not just its subsample. To do this, we use the method presented in Equation 9. Each method was run on six different three-dimensional grids with total sizes ranging from 51,200 to 219,700 grid points. We compare our fast inference method against five different subsample sizes for the naive method. Every experiment was run 10 times and the average time and RMSE is reported in Figure 2.

The grid-based Laplace's method vastly outperforms the naive method. Not only does the naive method get slower with an increase in grid size, but it also starts to perform worse, since it has to make predictions at a finer resolution from the same number of subsampled observations. Note that the naive method is also making predictions at roughly an order of magnitude fewer locations than the fast method because there are many grid points with zero observations.

**Comparison of likelihood functions.**  Next, we show in Figure 4 the importance of the wrapped normal likelihood when dealing with aliased data. We use the raw radial velocity data from 13 radar stations in the northeast US from the night of September 11, 2010. Figure 4(a) shows the inferred velocity field using our method with the Gaussian likelihood and Figure 4(b) shows the inferred velocity field using our method with the wrapped normal likelihood. Observe the region of the velocity field highlighted by the rectangle. The inference method with Gaussian likelihood fails to infer a reasonable velocity field in the presence of heavily aliased radial velocity measurements and has a substantially higher RMSE[1] than the method with the wrapped normal likelihood. The latter model correctly infers from raw aliased radial velocities that the birds over those stations are flying in the same general direction as birds over nearby stations.

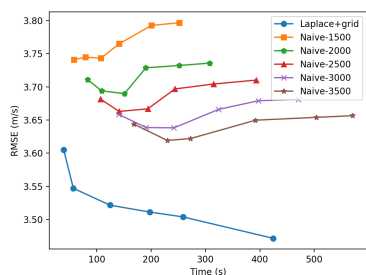

Figure 2: Time vs. RMSE of radial velocity measurements using six different methods for latent velocity inference.

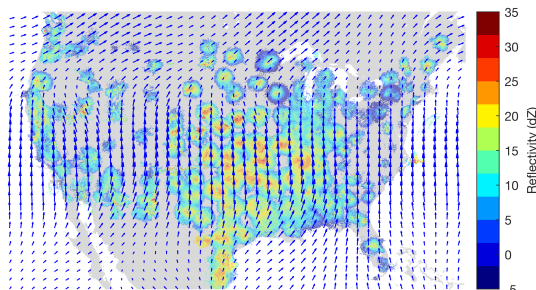

Figure 3: Density and velocity of bird migration on night of May 2, 2015. Northward migration occurs across the US, and is intense in the central US.

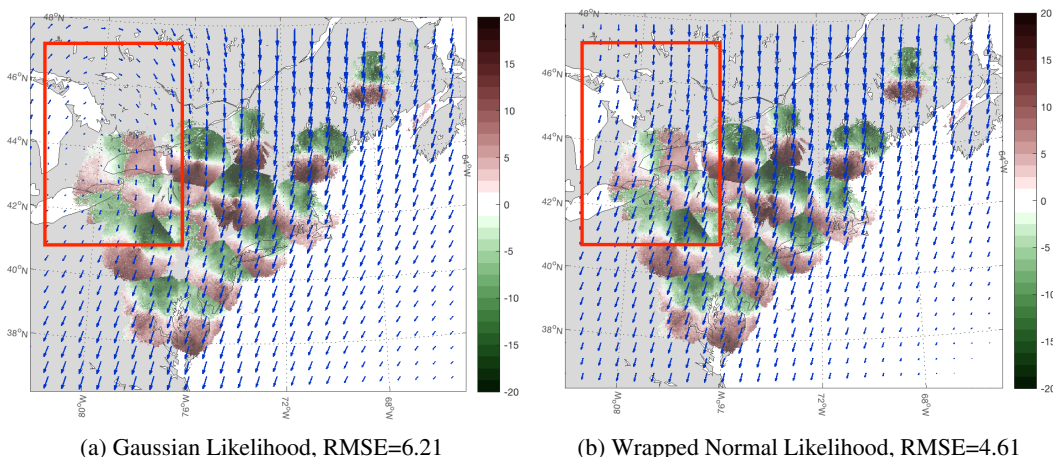

(a) Gaussian Likelihood, RMSE=6.21      (b) Wrapped Normal Likelihood, RMSE=4.61

Figure 4: Inference method performance using two likelihood functions on aliased data. Grid size is $100 \times 100 \times 9$; only the lowest elevation (500m above ground level) is displayed.

**Scaling to the continental US.** A unique aspect of our method is that it can, for the first time, assimilate data from all radar stations to reconstruct spatially detailed velocity fields across the whole US. An example is shown in Figure 1(d), which depicts northward bird migration on the night of May 2, 2015. The grid size is $240 \times 120 \times 10$; only the lowest elevation and every 5th velocity measurement is plotted. The reconstructed velocities can be combined with reflectivity data as shown in Figure 3 to observe both the density and velocity of migration. Future work can conduct quantitative analyses of migration biology using these measurements.

## 6 Conclusion and Future Work

We presented the first comprehensive solution to the problem of inferring latent velocities from radial velocity measurements from weather radar stations across the US. Our end-to-end method probabilistic model begins with raw radial velocity from many radar stations, and outputs valuable information about migration patterns of birds at scale. We presented a novel method to perform fast grid-based posterior inference even though our GP does not have a stationary kernel. The results of our methods can be used by ecologists to expand human knowledge about bird movements to advance conservation efforts and science.

Our current method is most suited to smooth velocity fields, such as those that occur during bird migration. A promising line of future work is to extend our techniques to infer wind velocity fields by measuring velocity of precipitation and wind-borne particles. We anticipate that our GP methodology

can also apply to this domain, but we will need to experiment with different kernels better suited to these velocity fields, which can be much more complex.

**Acknowledgments**

This material is based upon work supported by the National Science Foundation under Grant Nos. 1522054 and 1661259.

## Footnotes

[1]For aliased data, RMSE is measured between the observed value and the *closest alias* of the predicted value.

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
