[Reviews · NeurIPS 2018]

Reviewer 1



The paper proposes a method to compute the MAP velocity vector fields from radar line-of-sight velocity measurements, using a Gaussian process prior over the velocity fields. The method is demonstrated with bird migration data from 146 US radar stations. The paper is a good read and it nicely combines previous methods (linear projections from GP, fast-kernel matrix multiplication, Laplace’s method) to solve a real-world task. The method is validated experimentally with real data, yielding both a numerical as well as a visual improvement against a naive implementation. The idea of using Laplace's method even for Gaussian posteriors, to enable fast computation, might be useful also in other places, too. It would be a strong additional point, if one could argue that the new formulation would give a new, important real-world insight that was not known before. As it is, we get nice images, but it is not clear what the real-world relevance of these is. Some remarks and questions: -) Eq (1): "a_ij" is not defined here (though it is mentioned above) -) Eq line 143: Is there a "\sigma^2" missing? It appears again in Eq (6). -) L 144-145: I would say that $k_\theta$ is stationary, "Cov(y,y)" is not. The reason for non-stationarity of the latter is rather the direction vectors, not the locations of the stations. -) L 150: $[z;y]$ is Matlab notation, but not formal maths. Change? -) L 159: "number of points" refers to the number of measurements, not the number of stations. The number of stations is rather small (a couple 100s). The real computational problem is with the many measurements from each station. Clarify? -) Eq (10): $[x_i \in \Omega]$ ==> Better write "1_[x_i \in \Omega]"? -) L 204: return"s" -) L 205: from "M"? -) L 226: Why not all 146 available stations? I have read the author response. By novel real-world insight I meant a statement like "Birds fly nicely but then become irritated by Chicago O'Hare. They start flying too low and crash against trees." or sth. similar. But it is also ok, to keep such results to biology journals and conference. Anyway, a good paper.

Reviewer 2



-- Paper Summary The primary challenges posed by the problem considered in this paper arise from three main issues, namely 1) the data may be aliased, 2) only the radial velocity is measured (while the full velocity has to be inferred) and 3) the data is generally linked to a specific weather station. While some of these issues have been previously addressed, the paper delivers a solution which jointly tackles the three issues. This proposal builds upon the KISS-GP framework, while also developing an efficient way to carry out Laplace inference when incorporating a wrapped normal likelihood. While it was previously common practice to subsample the data, this approach can handle the full dataset (500k points) without compromising on speed. This is particularly important since the kernel is non-stationary, where standard inference techniques are not always applicable. -- Originality + Significance While applied papers do feature at NIPS, I believe that the theoretic contributions developed in this paper are only of minor significance to the general community, which tends to value significant algorithmic contributions above advances in applied areas. The generality of the proposed improvements to existing GP methods also lose their impact due to the paper’s focus on a single dataset. While there is some emphasis on the Laplace method being used to carry out exact inference, the authors then claim that this is no longer possible when incorporating the wrapped normal likelihood. The novelty of the Laplace computation is also unclear to me. Similar work was carried out in relation to grid-structured inputs and conjugate gradient for Laplace in ‘Fast Kronecker Inference in Gaussian Processes with non-Gaussian likelihoods (Flaxman et al, 2015)’ and ‘Preconditioning Kernel Matrices (Cutajar et al, 2016)’, but these papers are not discussed or cited here. The overall scope of this work is also limited to a particular problem and its associated data. While the conclusion indicates that there may in fact be other similar problems which are worth investigating (e.g. inferring wind velocity from precipitation or wind particles), it is not immediately clear whether the model developed in this paper can be similarly successful in these other instances. -- Quality The derivations put forward in the paper appear to be correct, drawing from results which were previously established in the literature on structured kernel interpolation for Gaussian processes. The superior performance of the method is not altogether surprising given the success of its intermediate components showcased in other papers. The impact of using pseudo-observations in order to fill up incomplete grids should however be elaborated by at least re-iterating the motivations given in the original KISS-GP paper. The results obtained using the proposed method are very promising in comparison with other naïve GP approaches. The running time comparison is also very useful in validating the authors’ intuition that the increase in predictive performance does not incur a penalty in speed. There is however very little context for how successful the results are beyond the narrow GP setting. The “6 different methods” highlighted in L234 are only the proposed method and the naïve GP method with 5 different training set sizes. As alluded to before, the performance improvements enabled by this proposed method are expected given their success in other domains. Although the temporal aspect of this problem is not considered in the paper (even though it is implied in the introductory comment that such analysis could answer questions “over more than two decades”), a brief investigation into INLA methods may also be warranted. Scalable deep GPs may also be useful for modelling non-stationarity behaviour. Finally, while I understand that such data may not be readily available, I think it would be beneficial to have additional results for the model on similarly structured datasets having similar aliasing issues for example. I would expect similarly good results on other such spatio-temporal problems, but it would be best if this could also be supplemented with actual experiments. -- Writing and Clarity I cannot fault the writing of this paper – it is extremely well written with nary a typo to be found (I only spotted L62). The problem is adequately set up, and the individual components of the model design are well-motivated and described in an orderly manner. The paper is also sufficiently detailed to properly understand without having to rely on the supplementary material. The notation is easy to follow and interpret. -- Overall Recommendation I appreciate that having an end-to-end scalable model which takes partial measurements as input and successfully models actual wind velocities is useful for specific problems and datasets in this particular domain. This is evidenced by the model’s success compared to standard GP models. Nevertheless, even though the problem is indeed interesting and the paper is in a state fit for publication, I do not think NIPS is the ideal venue unless a wider variety of problems or data settings are explored. Essentially, the minor theoretic contributions must be compensated for by a wider and more comprehensive set of experiments. ------------- -- Post-rebuttal I thank the authors for replying to my comments. After reading the rebuttal and other reviews, I have bumped up my score from 4 to 6 in order to better reflect its quality as an application track paper. Nonetheless, for other reasons mentioned in the full-review, I do not feel inclined to argue more strongly for its acceptance.

Reviewer 3



This paper proposes and validates an approach for inferring latest velocities given weather radar data. The proposed approach is a framework based on Gaussian Process (GP) models, and addresses many aspects (partial velocity measurements, station specific geometries, measurement noise, and other domain specific issues, such as "aliasing", etc.) in a unified manner. The empirical application is done on the problem of inferring bird migration patterns/speeds given the weather radar data. Sophisticated and talented use of various existing techniques are made to solve this problem elegantly and satisfactorily. Only minuses of this paper are: significance and applicability of the proposed approach to other application areas are less clear, and so is the significance of the proposal on the methodological development of central interest to the NIPS community. Nonetheless, this is a good application paper, which probably deserves to be represented in the conference.